# Examining workweek variations in computer usage patterns: An application of ergonomic monitoring software

**Taehyun Roh**[1], **Chukwuemeka Esomonu**[2], **Joseph Hendricks**[2], **Anisha Aggarwal**[3], **Nishat Tasnim Hasan**[1], **Mark Benden**[2]*

**1** Department of Epidemiology and Biostatistics, School of Public Health, Texas A&M University, College Station, Texas, United States of America, **2** Department of Environmental and Occupational Health, School of Public Health, Texas A&M University, College Station, Texas, United States of America, **3** Department of Health Behavior, School of Public Health, Texas A&M University, College Station, Texas, United States of America

* mbenden@tamu.edu

**Data Availability Statement:** Data cannot be shared publicly because of privacy and confidentiality. The data are available from the Institutional Review Board of Texas A&M

## Abstract

Alternative work arrangements have emerged as potential solutions to enhance productivity and work-life balance. However, accurate and objective measurement of work patterns is essential to make decisions about adjusting work arrangements. This study aimed at evaluating objective computer usage metrics as a proxy for productivity using RSIGuard, an ergonomics monitoring software. Data were collected from 789 office-based employees over a two-year period between January 1, 2017 and December 31, 2018 at a large energy company in Texas. A generalized mixed-effects model was utilized to compare computer usage patterns across different days of the week and times of the day. Our findings demonstrate that computer output metrics significantly decrease on Fridays compared to other weekdays, even after controlling for total active hours. Additionally, we found that workers' output varied depending on the time of day, with reduced computer usage observed in the afternoons and a significant decrease on Friday afternoons. The decrease in the number of typos was much less than that in the number of words typed, indicating reduced work efficiency on Friday afternoons. These objective indicators provide a novel approach to evaluating the productivity during the workweek and can help optimize work arrangements to promote sustainability for the benefit of employers, employees, and the environment.

## Introduction

The 21st century has undergone a tremendous transformation, bringing about the need for alternative work arrangements to better align with the changing needs of employers and employees [1]. However, there is a need to identify the benefits and drawbacks of these arrangements to inform policies and practices for the modern-day workforce. The compressed workweek and hybrid work are two common alternative work arrangements that have emerged as potential solutions to promote productivity and work-life balance. The compressed workweek involves compressing the standard 40-hour workweek into fewer than five days, with employees

University for researchers who meet the criteria for access to confidential data. The contact for such data access is Kelly Drake, Human Research Protection Program Coordinator via email (kdrake@tamu.edu) or phone (979-458-2557).

**Funding:** This research was funded by the Texas A&M University Ergonomics Center. The funders had no role in study design, data collection and analysis, decision to publish, or preparation of the manuscript.

**Competing interests:** The authors have declared that no competing interests exist.

working longer hours each day [2]. On the other hand, hybrid work allows employees work at both a traditional office and home workspace, offering greater flexibility [3].

Promoting workforce sustainability and employee well-being are critical considerations for businesses seeking long-term success. Studies have shown that alternative work arrangements have been implemented in many workplaces to promote employee's work-life balance [4]. For example, work-life-friendly programs, such as flexible working hours, parental leave entitlements, and childcare facilities, have been implemented in many workplaces. Work-life balance is directly linked to organizational and social sustainability, and many developed nations' governments have called for strategies to achieve a better balance between work and life [5, 6]. Other countries and companies are exploring the impacts of alternative work arrangements, such as the 4-day workweek in Spain, Unilever in New Zealand, Shopify in Canada, and Microsoft in Japan [7–10]. It is, therefore, essential to examine the benefits and drawbacks of alternative work arrangements in greater detail to inform policies and practices for the modern-day workforce.

Our study highlights the importance of using accurate and objective quantitative measures for evaluating work performance and assessing the impact of optimal work arrangements on productivity. Especially, this study aims to fill the gap in existing literature by examining variations in work patterns across days of the week using daily computer usage metrics of office-based workers. Traditional methods such as diaries, self-report surveys, and supervisory performance appraisals are subjective and subject to various biases, mood, and invasion of privacy [11, 12]. Employers may also be subject to bias in noting benefits to the company in survey responses [13]. While objective measurements of work productivity can be obtained through the use of computers or wearable devices, wearable devices can be invasive and raise concerns about privacy and user comfort, as they require multiple sensor devices to be attached to users [14]. In contrast, non-invasive computer usage metrics such as typing speed, typing errors, interkey interval, backspace use, and mouse activities have been suggested and utilized in previous studies to evaluate mental distress and work productivity objectively [15–17]. The previous study provided evidence supporting the accuracy of measuring computer use directly, which helps to minimize inaccuracies arising from self-reported data [18]. These metrics provides non-invasive and objective measures of productivity, allowing researchers to develop a more com-prehensive understanding of behaviours and performance patterns in the workplace.

The main objective of this study is to determine variations in work patterns across days of the week using objective data from daily computer usage metrics of office-based workers in the corporate arm of an energy company in Houston, Texas, and its application to assess computer output, which we consider a proxy for overall work productivity. This paper presents key findings on computer use metrics throughout the workweek, explores psychological and behavioral explanations for patterns, and discusses implications for sustainability and organizational management.

## Materials and methods

### Study design and participants

The study recruited 781 office-based workers from the Houston campus of a large international oil and gas corporation. The study included participants from various computer-oriented, office-based positions, such as admins, geologist, accountants, and engineers, within the company. Although demographic and other individual factors were not collected, these characteristics were assumed to be distributed evenly, and the effects of outliers would be minimal,

due to the large sample size. Additionally, the participant age range was assumed that of the average working adult population.

A fully within-subjects design used day of the week as the independent variable (Monday–Friday) and computer output metrics as the dependent variables. By examining M-F changes within participants over a two-year period, this study aimed to control for artifacts such as activity that may not strongly contribute to overall productivity, as these are likely stable fluctuations within-person over time. This design could also control for differences between persons with regard to job types and duties.

This research complied with the tenets of the Declaration of Helsinki, and approved by the Institutional Review Board of the Texas A&M University (IRB2018-1623M). Written informed consent was obtained from all participants.

## Measurements

Computer usage metrics were collected using RSIGuard software version 6 (Cority Enviance, Carlsbad, CA, USA), which was installed on employees' computers. This software has previously been used and validated to measure computer usage and productivity [19–21]. Table 1 presents the list of the metrics measured by RSIGuard software.

The dataset included 130,681 observations from 789 individuals, representing daily inputs from participants between January 1, 2017, and December 31, 2018. Data from the period between August 1 and September 30, 2017, were excluded due to the impact of Hurricane Harvey on participants' work during this period. After excluding observations with zero recorded total active hours, keyboard or mouse hours, as well as those with zero or missing values in the total number of words typed, mouse distances, and mouse clicks and scrolls, we further excluded observations with disparities between the overall number of words typed or typos and the sum of numbers of words typed or typos in the morning and afternoon. In total, 111,719 observations met the inclusion criteria and were used for our analysis.

## Statistical analysis

A generalized mixed-effects model was used to analyze the data with repeated, longitudinal, and correlated measures [22]. All computer metrics data showed a right-skewed distribution

**Table 1. List of computer metrics measured by RSIGuard software.**

| Metrics | Description |
|---|---|
| Active Hours | Total active time spent actively using at the computer (keyboard, mouse, or both) |
| Mouse Hours | Total hours spent using the mouse |
| Keyboard Hours | Total hours spent using the keyboard |
| Words Typed | Total number of words typed per day; any series of letters followed by a space were considered a new word |
| Words Typed AM | Total words typed in the morning |
| Words Typed PM | Total words typed in the afternoon |
| Typos | Number of errors of spells made while typing words |
| Typos AM | Number of errors made while typing words in the morning |
| Typos PM | Number of errors made while typing words in the afternoon |
| Mouse Clicks | Total number of mouse clicks |
| Mouse Distance | Total number of pixels covered by the mouse cursor |
| Mouse Scrolls | Number of mouse scrolls |

**Table 2. Least squared geometric means (95% CIs) of computer metrics by day after adjusting for active hours.**

|  | Monday | Tuesday | Wednesday | Thursday | Friday |
|---|---|---|---|---|---|
| **Total Words Typed** | 427.0 (405.0, 450.3) | 442.3 (419.5, 466.3) | 444.6 (421.7, 468.8) | 443.2 (420.3, 467.3) | 372.2 (352.9, 392.6) |
| **Typos** | 145.7 (142.1, 149.4) | 140.9 (137.4, 144.4) | 140.3 (136.9, 143.8) | 140.9 (137.5, 144.5) | 138.6 (135.2, 142.2) |
| **Mouse Distance (x100)** | 1064.4 (1035.1, 1094.5) | 1075.1 (1045.5, 1105.5) | 1080.6 (1050.8, 1111.1) | 1098.6 (1068.4, 1129.7) | 991.4 (963.9, 1019.8) |
| **Mouse Clicks** | 1826.2 (1781.3, 1872.4) | 1846.2 (1800.8, 1893.0) | 1853.8 (1808.0, 1900.6) | 1876.2 (1830.1, 1923.5) | 1771.9 (1728.0, 1817.1) |
| **Mouse Scrolls** | 1196.4 (1066.6, 1342.2) | 1191.4 (1062.1, 1336.6) | 1209.9 (1078.6, 1357.4) | 1219.3 (1086.9, 1367.9) | 1161.5 (1035.0, 1303.6) |

and were natural log-transformed before being included in the model [23]. The primary independent variable across all analyses was the day of the week, and separate models were built for each dependent variable (e.g., words typed). The total number of words typed was included in the model to analyze the pattern of the number of typos for adjustment. For those data that included time of day (i.e., AM vs. PM), time of day was included as an additional independent variable in the models, and its interaction with day of the week was analyzed. In additional analyses, total mouse and keyboard computer activity was controlled for a more rigorous hypothesis testing. Post hoc pairwise multiple comparisons for each day of the week and time of day were conducted using Tukey's method. The least squared geometric means (LSGM) of the computer metrics, which are predicted geometric mean values of the dependent variable by day from the regression model, were calculated. The data were completely anonymous, so demographic and other individual factors were not collected. However, due to the large sample size, it is assumed that those characteristics would be distributed evenly, and the effects of outliers would be minimal. The participant age range is also assumed to be that of the average working adult population. All statistical analyses were conducted with SAS software (version 9.4; SAS Institute, Inc., Cary, North Carolina). The statistical significance was declared at $p < 0.01$.

## Results

In Table 2, the LSGM of computer metrics measured by RSIGuard by day is presented after adjusting for active hours, and in Table 3, the percent changes in these metrics between each pair of weekdays are shown. Increasing trends over the week were observed for most metrics,

**Table 3. Percent changes[a] (95% CIs) in computer metrics between each pair of weekdays.**

| Comparison | Words Typed | Typos | Mouse Distance | Mouse Clicks | Mouse Scroll |
|---|---|---|---|---|---|
| **Mon—Tue** | 3.45 (1.00, 5.84)* | -3.44 (-4.36, -2.54)* | 0.99 (-0.61, 2.58) | 1.08 (-0.34, 2.49) | -0.42 (-3.93, 2.97) |
| **Mon—Wed** | 3.95 (1.51, 6.33)* | -3.85 (-4.77, -2.95)* | 1.50 (-0.10, 3.07) | 1.48 (0.06, 2.88) * | 1.11 (-2.35, 4.46) |
| **Mon—Thu** | 3.64 (1.19, 6.03)* | -3.38 (-4.29, -2.47)* | 3.11 (1.54, 4.67)* | 2.66 (1.25, 4.04)* | 1.87 (-1.58, 5.20) |
| **Mon—Fri** | -14.7 (-17.8, -11.7)* | -5.08 (-6.07, -4.11)* | -7.36 (-9.19, -5.56)* | -3.07 (-4.63, -1.53)* | -3.00 (-6.85, 0.71) |
| **Tue—Wed** | 0.52 (-1.99, 2.97) | -0.40 (-1.28, 0.47) | 0.51 (-1.10, 2.09) | 0.40 (-1.03, 1.81) | 1.53 (-1.90, 4.84) |
| **Tue—Thu** | 0.20 (-2.33, 2.66) | 0.06 (-0.82, 0.93) | 2.14 (0.55, 3.70)* | 1.59 (0.18, 2.99)* | 2.28 (-1.13, 5.58) |
| **Tue—Fri** | -18.8 (-22.0, -15.7)* | -1.59 (-2.53, -0.65)* | -8.44 (-10.3, -6.63)* | -4.20 (-5.77, -2.65)* | -2.57 (-6.38, 1.10) |
| **Wed—Thu** | -0.32 (-2.86, 2.15) | 0.46 (-0.42, 1.32) | 1.64 (0.04, 3.21)* | 1.20 (-0.22, 2.60) | 0.77 (-2.70, 4.12) |
| **Wed—Fri** | -19.5 (-22.6, -16.4)* | -1.18 (-2.13, -0.25)* | -8.99 (-10.8, -7.18)* | -4.62 (-6.19, -3.07)* | -4.16 (-8.03, -0.43)* |
| **Thu—Fri** | -19.1 (-22.2, -16.0)* | -1.65 (-2.60, -0.71)* | -10.8 (-12.7, -8.96)* | -5.88 (-7.48, -4.31)* | -4.97 (-8.87, -1.21)* |

[a] A percent change means the percent difference in each metric between two days. For example, the number of words typed increased by 3.45% between Mondays and Tuesdays, and decreased by 19.1% between Thursdays and Fridays.

* Statistically significant different at $p < 0.01$ after Tukey's pairwise comparisons.

followed by statistically significant reductions on Fridays. The mean for total words typed was 427 (95% CI 405.0, 450.3) on Monday, with a significant increase by 3.45% (95% CI 1.00, 5.84) on Tuesday and no significant changes until Thursday. A significant decline was observed on Friday by 19.1% (95% CI -22.2, -16.0), with the lowest LSGM (372.2, 95% CI 352.9, 392.6) compared to other days of the week. Total typing errors (typos) showed a significant decrease between Monday and Tuesday by 3.44% (95% CI -4.36, -2.54) after adjusting for the active hours and total number of words typed, followed by a non-significant decrease until Thursday and a significant decrease on Friday by 1.65% (95% CI -2.60, -0.71). Similar findings were noted with all mouse metrics, including the total distance covered by the mouse cursor and the total number of scrolls, with significant drops on Friday. Steady increases were also found with the mouse clicks and scrolls, with a statistically significant drop on Friday compared to other days of the week.

After the duration of active use was adjusted, all computer usage metrics were found to be significantly lower on Friday compared to other weekdays, as shown in Fig 1, which presents the LSGMs and 95% CIs of computer metrics.

Furthermore, the interaction effects of the day of the week with time of day (AM/PM) on the numbers of words typed and typos were examined (Fig 2). Despite adjusting for the day by AM/PM interaction term, a significant main effect of the day of the week was still observed for

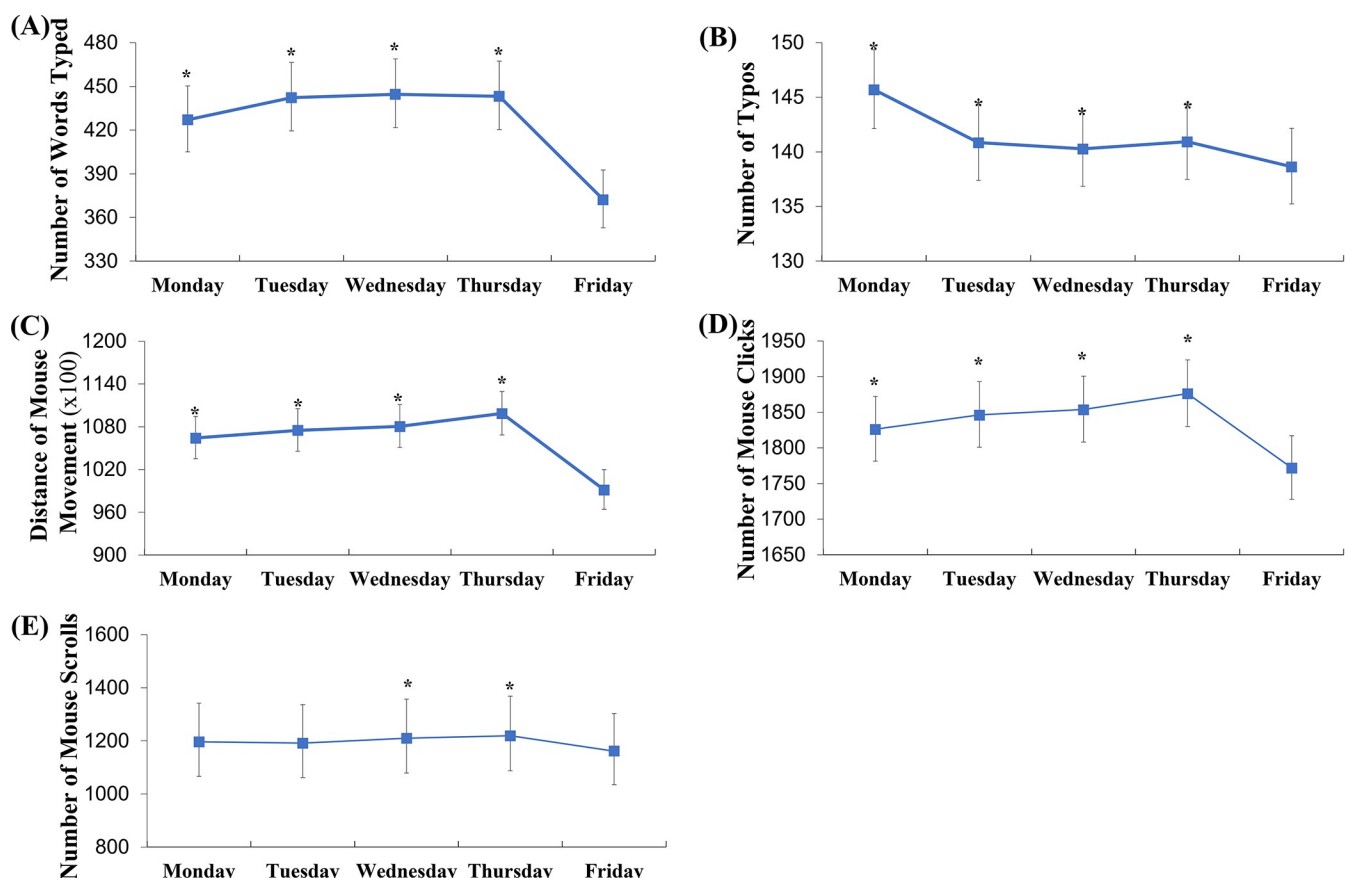

**Fig 1. Comparison of LSGMs of computer metrics on other weekdays versus Fridays.** (A) number of words typed, (B) number of typos, (C) distance of mouse cursor movements, (D) numbers of mouse clicks, and (E) number of mouse scrolls. All estimates were adjusted for active working hours. For the typos, the number of words typed was included additionally for adjustment. Each marker represents the least-squared geometric mean of each variable with 95% CI. * Statistically significant at p < 0.01, compared to Fridays.

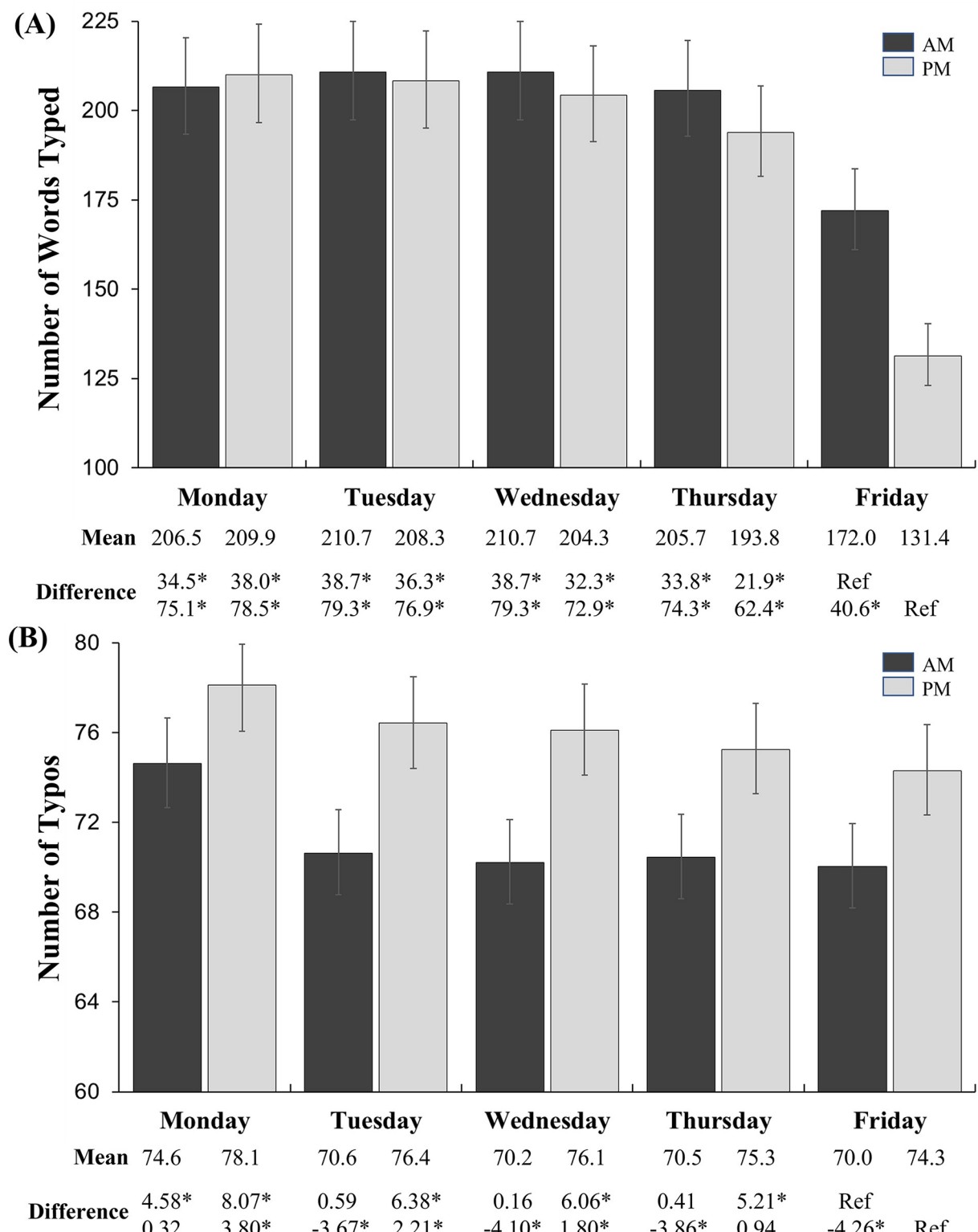

**Fig 2. The interaction effect of day of the week and time of day on computer metrics.** (A) numbers of words typed and (B) number of typos. For the analysis of typos, the number of words typed was included additionally for adjustment. Each column represents the least-squared geometric mean of each variable with 95% CI. * Statistically significant at p < 0.01.

total words typed per day (p < 0.01). Consistent with the above analysis, the pairwise post-hoc test revealed a significant decrease in the number of words typed on Friday, particularly in the afternoon. For typos, there was a significant day of the week by time-of-day interaction effect (p < 0.01). However, the trends of typos adjusted for the number of words typed were slightly different. The results showed that, on average, people made significantly more typos in the afternoon than in the morning (mean difference 4.86) across all days of the week, after adjusting for the number of words typed. Notably, there was no difference in the means of numbers of words typed between Tuesday and Friday in the mornings, but in the afternoons, the number of typos showed gradual reductions through Thursday, followed by leveling off on Fridays. This contrasts with the significant drop in the number of words typed on Friday afternoons.

## Discussion

The present study analyzed workplace computer usage across the workweek by examining multiple observations of computer output metrics from 789 participants over two years. We observed that computer use metrics measured by RSIGuard (total words typed, total mouse distance, mouse clicks, and scrolls) gradually increased throughout the workweek, with the greatest reduction observed on Friday afternoons. Interestingly, the number of typos only decreased by 1.65%, significantly fewer than the 19.1% decrease in the number of words typed on Friday. Our analysis also revealed a significant interaction effect between the day of the week and time of day (AM/PM), with employees being less active in the afternoons and the average number of typos being higher in the afternoons, especially on Fridays.

A major strength of this study is the use of objective and quantitative measures of computer usage metrics to analyze workweek variations in a large sample over an extended period. This provides a basis for future research to explore the causal relationship between computer usage and workplace productivity. Previous studies have shown that using computer metrics reduces inaccuracies from self-reports and other subjective methods, providing a better understanding of workplace behavior [11, 18]. Our findings can be leveraged by business leaders to review current work arrangements and ensure optimal use of the most productive days in the workweek, given that computer metrics have been previously used as performance indicators [16, 21].

Our findings align with a previous study, which found that employees' productivity, as measured by the number of work tasks completed, steadily increased from Monday through Wednesday, followed by a decrease on Thursday and Friday [24]. This trend may be due to the higher perceived stress workers face on Monday, often referred to as "Monday Blues," as they return to work after the weekend, and experience lower mood and a lack of motivation, requiring time to reorient themselves [25, 26]. Additionally, a study has shown that workers' cortisol levels are higher on Monday mornings, indicating the negative impact of anticipating an upcoming workday on stress [27]. However, as workers gain more experience over the course of workweek, they tend to become more proficient in their work, resulting in higher productivity at the beginning of the week, with the peak of productivity occurring in the middle of the week [25].

Employees may experience mounting stress and mental and physiological fatigue as the workweek progresses. This finding is consistent with the study by Riley et al., which reported a decline in cognitive and attentional abilities in the afternoon and a corresponding increase in errors throughout the day [28]. Although employees may partially recover from mental fatigue and stress from mental detachment, such as exercise and sleep in the evening, previous studies have shown that this recovery is limited, and their mental resources do not fully restore, accumulating over the workweek [29]. As a result, employees report higher levels of mental fatigue, stress, and negative mood on Thursdays and Fridays, compared to Mondays and Tuesdays

[30]. Other studies showed that workers' self-reported job satisfaction, engagement, and productivity levels were lowest on Fridays [31, 32]. These findings suggest that the traditional five-day workweek may not be the most effective model for promoting employee well-being and productivity.

Our study provides insight into the impact of alternative work arrangements on sustainability. Flexible work arrangements, such as hybrid work or a four-day workweek, may help mitigate the negative effects of long workweeks and promote better employee well-being and productivity. For instance, employers may be more open to allowing employees to telecommute on Fridays or implementing shortened workweeks with Fridays off. Research has consistently shown that remote work or working from home can positively impact employee well-being, job satisfaction, and productivity. Studies have found that remote work reduces stress associated with commuting and workplace politics and provides employees with greater control over their work schedule, which helps reduce mental fatigue and burnout and increase job satisfaction [33, 34].

Moreover, remote work offers flexibility, reducing work-family conflicts and improving family relationships [35, 36]. It also allows for increased time for leisure and exercise, which positively impacts both physical and mental health and increases work productivity [37]. In addition to the benefits for employees, remote work has environ-mental advantages. It reduces transportation fuel consumption, $CO_2$ emissions, and other pollutants, reducing noise pollution and saving energy and material resources [38, 39]. A study estimated the impact of working from home on greenhouse gas emissions and associated energy costs in the residential, commercial, and transportation sectors [40]. The results showed that working from home for 1.5 days per week reduces annual carbon dioxide emissions by about 1.21 million metric tons and reduces annual transportation costs for an employee by 30%.

In addition, numerous studies reported that compressed workweeks or reduced working hours could also contribute to substantial progress in employee, employer, and environmental sustainability. A study found that adopting a compressed work schedule (CWS) led to a higher job, life, and leisure time satisfaction for both employees and employers. A study found that adopting a CWS led to higher job, life, and leisure time satisfaction for both employees and employers [41]. Furthermore, recent studies have shown that implementing a CWS can lead to increased productivity and decreased electricity use. For instance, Microsoft Japan tested a CWS among a 2300-person workforce for five weeks and found a 40% increase in productivity and a 23% decrease in electricity use [7]. A report estimated a 10% reduction in working hours was predicted to cause declines in ecological footprint, carbon footprint, and carbon dioxide emissions by 12.1%, 14.6%, and 4.2%, respectively, based on the data from 29 nations between 1970 and 2007 [42]. According to a recent report on the experiences of 61 UK companies and 2,900 employees [43], a four-day work week can improve employee health by reducing work-related stress and burnout. The companies were allowed to set their own schedules as long as they reduced working hours meaningfully without reducing pay. During the six-month pilot period, employees worked an average of 34 hours per week, down from 38 hours, and employers saw benefits such as increased revenue and reduced employee turnover and absenteeism. Another report projected that the UK's carbon footprint could be reduced by 21.3% by the year 2025 if a four-day workweek, with no reduction in pay, is implemented, which is greater than the total carbon footprint of Switzerland [44].

However, this study has several limitations. Firstly, the data were collected from white-collar workers in a single corporate energy firm, and it may not be appropriate to generalize these findings to other fields of work. Secondly, other activities not observed in this study could significantly impact overall productivity. For example, the reduction in computer use might be caused by engaging in different types of activities, such as participating in meetings or

planning on Friday. However, those activities can be conducted remotely, which cannot override the necessity of alternative work arrangements and their positive impacts on sustainability. Finally, information on demographic, behavior, and other individual characteristics beyond job type was not collected, which limits the study's ability to provide a holistic view of other factors that may have influenced the observed changes. Therefore, more research is needed to confirm our findings and determine the generalizability of our findings across different industries and job types.

## Conclusions

This study's emphasis on the importance of objective and quantitative measures of computer usage metrics is particularly noteworthy in the context of the modern workplace, where the use of technology is becoming increasingly prevalent. The use of these measures can provide a more accurate assessment of work performance and allow for the identification of opportunities to improve productivity. The study's findings suggest that fatigue and stress can accumulate throughout the workweek, potentially leading to decreased productivity, particularly on Fridays. However, the study also offers suggestions for mitigating these effects by implementing measures to provide opportunities for breaks and rest. These measures may include flexible work arrangements, such as telecommuting on Fridays or shortened workweeks, which can promote employee health, work-life balance, and productivity. The adoption of alternative work arrangements can offer significant long-term benefits to businesses, including increased employee satisfaction, reduced absenteeism, and increased productivity. Furthermore, these arrangements can contribute to environmental sustainability by reducing transportation fuel consumption, $CO_2$ emissions, and other pollutants. Despite the study's limitations, such as the narrow scope of participants and the inability to control for all factors that may impact productivity, the findings suggest that businesses can benefit from using objective and quantitative measures of computer usage metrics to assess work performance and identify opportunities to improve productivity. Future research can build on these findings to explore the potential of alternative work arrangements further and identify strategies to optimize work performance and promote sustainability in the workplace. In conclusion, this study underscores the importance of objective and quantitative measures of computer usage metrics in assessing work performance and identifying opportunities for improvement. The findings highlight the potential benefits of implementing measures to mitigate fatigue and promote rest, including alternative work arrangements, such as telecommuting on Fridays or shortened workweeks. These arrangements can promote employee health, work-life balance, and productivity, all critical components of sustainability in a business setting. Further research is needed to confirm the study's findings and identify strategies to optimize work performance and promote sustainability across different industries and job types.

## Author Contributions

**Conceptualization:** Chukwuemeka Esomonu, Mark Benden.

**Formal analysis:** Taehyun Roh, Joseph Hendricks.

**Funding acquisition:** Mark Benden.

**Methodology:** Taehyun Roh, Joseph Hendricks.

**Supervision:** Taehyun Roh, Mark Benden.

**Visualization:** Taehyun Roh, Nishat Tasnim Hasan.

**Writing – original draft:** Taehyun Roh, Chukwuemeka Esomonu, Joseph Hendricks, Anisha Aggarwal.

**Writing – review & editing:** Taehyun Roh, Joseph Hendricks, Anisha Aggarwal, Mark Benden.

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
