## [Decision Letter · Decision Letter 0]

2 May 2023

PONE-D-23-09730Examining Workweek Variations in Computer Usage Patterns: An Application of Ergonomic Monitoring SoftwarePLOS ONE

Dear Dr. Benden,

Thank you for submitting your manuscript to PLOS ONE. After careful consideration, we feel that it has merit but does not fully meet PLOS ONE’s publication criteria as it currently stands. Therefore, we invite you to submit a revised version of the manuscript that addresses the points raised during the review process.

We look forward to receiving your revised manuscript.

Kind regards,

Radoslaw Wolniak, full professor

Academic Editor

PLOS ONE

Journal Requirements:

   "This research was funded by the Texas A&M University Ergonomics Center."

Additional Editor Comments:

The Authors should adjust the paper according to the Reviewers comments.

Reviewers' comments:

Reviewer's Responses to Questions

**Comments to the Author**

1. Is the manuscript technically sound, and do the data support the conclusions?

Reviewer #1: Yes

Reviewer #2: Yes

2. Has the statistical analysis been performed appropriately and rigorously? 

Reviewer #1: Yes

Reviewer #2: Yes

3. Have the authors made all data underlying the findings in their manuscript fully available?

Reviewer #1: Yes

Reviewer #2: No

4. Is the manuscript presented in an intelligible fashion and written in standard English?

Reviewer #1: Yes

Reviewer #2: Yes

5. Review Comments to the Author

Reviewer #1: The article is factually correct. The article concerns current scientific issues. The study assessed computer use as an indicator of performance. The data was collected from 789 office workers over two years at a major Texas energy company. The analysis and conclusions are correct. The literature is selected correctly. Graphic elements complement the content and are made carefully. The tables are legible and understandable. I don't feel qualified to judge about the English language and style. In my opinion, the article can be published.

Reviewer #2: A manuscript with a very interesting title; however, the work needs some improvements. In the abstract, I recommend adding when exactly the research was conducted.

Introduction. I recommend highlighting what is novel in the work and what gap is filled by the work. At this point, I also suggest briefly describing how the manuscript is organized.

Methodology. In this section, respondents require a more robust characterization. I recommend adding information about their age, gender, work experience, how they were selected for the study, and what exactly their office work consisted of. On line 93, the author writes that: "Additionally, we assumed that." I don't quite understand why the plural is used since the author is the sole author of the work.

Conclusion: very short, too general, needs a lot of refinement.

I also recommend adding more 2022 and 2023 literature.

6. PLOS authors have the option to publish the peer review history of their article (what does this mean?). If published, this will include your full peer review and any attached files.

Reviewer #1: No

Reviewer #2: No

---

## [Author Response · Author response to Decision Letter 0]

7 Jun 2023

Response to Reviewers

Manuscript ID: PONE-D-23-09730 

Manuscript Title: Examining Workweek Variations in Computer Usage Patterns: An Application of Ergonomic Monitoring Software

Response to Reviewer 1’s Comments:

Reviewer #1: The article is factually correct. The article concerns current scientific issues. The study assessed computer use as an indicator of performance. The data was collected from 789 office workers over two years at a major Texas energy company. The analysis and conclusions are correct. The literature is selected correctly. Graphic elements complement the content and are made carefully. The tables are legible and understandable. I don't feel qualified to judge about the English language and style. In my opinion, the article can be published.

Response: Thank you for your comments and feedback on our manuscript. 

Response to Reviewer 2’s Comments:

Reviewer #2: A manuscript with a very interesting title; however, the work needs some improvements. 

1. In the abstract, I recommend adding when exactly the research was conducted.

Response: We have now included the specific time period (January 1, 2017 - December 31, 2018) during which data was collected in the abstract.

2. Introduction. I recommend highlighting what is novel in the work and what gap is filled by the work. At this point, I also suggest briefly describing how the manuscript is organized.

Response: We have updated the Introduction section as recommended by the reviewer. In addition, we have included a brief description of the organization of the manuscript at the end of the Introduction.

3. Methodology. In this section, respondents require a more robust characterization. I recommend adding information about their age, gender, work experience, how they were selected for the study, and what exactly their office work consisted of. 

Response: This pilot study aims to evaluate the effectiveness of ergonomic software in assessing performance trends. Unfortunately, we were unable to collect detailed individual-level data such as age, gender, and work experience. We have included this information as a limitation of our study in the discussion section. Despite this limitation, our preliminary data has shown promising results, and we plan to expand our study by collecting more data from participants and conducting a more comprehensive analysis as the reviewer recommended.

4. On line 93, the author writes that: "Additionally, we assumed that." I don't quite understand why the plural is used since the author is the sole author of the work.

Response: We used the plural form in this study because it was conducted by six scholars. However, we have revised our writing to use the passive voice instead of active voice.

5. Conclusion: very short, too general, needs a lot of refinement.

Response: I agree with the reviewer's comment, and we have taken their feedback into consideration by elaborating and refining the conclusion section further. 

6. I also recommend adding more 2022 and 2023 literature.

Response: We added more recent literature in the revision (now, 14 references published in 2022 and 2023, and 14 references published in 2020 and 2021. 

Thank you for your comments and feedback on our manuscript. We want to thank you for the time and effort you put into reviewing this manuscript. Your comments have helped contribute to improving the quality of the manuscript.

---

## [Decision Letter · Decision Letter 1]

19 Jun 2023

Examining Workweek Variations in Computer Usage Patterns: An Application of Ergonomic Monitoring Software

PONE-D-23-09730R1

Dear Dr. Benden,

We’re pleased to inform you that your manuscript has been judged scientifically suitable for publication and will be formally accepted for publication once it meets all outstanding technical requirements.

Kind regards,

Radoslaw Wolniak, full professor

Academic Editor

PLOS ONE

Additional Editor Comments (optional):

Reviewers' comments:

Reviewer's Responses to Questions

**Comments to the Author**

1. If the authors have adequately addressed your comments raised in a previous round of review and you feel that this manuscript is now acceptable for publication, you may indicate that here to bypass the “Comments to the Author” section, enter your conflict of interest statement in the “Confidential to Editor” section, and submit your "Accept" recommendation.

Reviewer #2: All comments have been addressed

2. Is the manuscript technically sound, and do the data support the conclusions?

Reviewer #2: Yes

3. Has the statistical analysis been performed appropriately and rigorously? 

Reviewer #2: Yes

4. Have the authors made all data underlying the findings in their manuscript fully available?

Reviewer #2: Yes

5. Is the manuscript presented in an intelligible fashion and written in standard English?

Reviewer #2: Yes

6. Review Comments to the Author

Reviewer #2: The manuscript has been improved in accordance with recommendations.The authors improved both the abstract, methodology, and conclusion sections. The manuscript looks legible. Thank you.

7. PLOS authors have the option to publish the peer review history of their article (what does this mean?). If published, this will include your full peer review and any attached files.

Reviewer #2: No

---

## [Editor Report · Acceptance letter]

27 Jun 2023

PONE-D-23-09730R1 

Examining Workweek Variations in Computer Usage Patterns: An Application of Ergonomic Monitoring Software 

Dear Dr. Benden:

I'm pleased to inform you that your manuscript has been deemed suitable for publication in PLOS ONE. Congratulations! Your manuscript is now with our production department. 

Kind regards, 

on behalf of

Professor Radoslaw Wolniak 

Academic Editor

PLOS ONE